# Undiagnosed hypertension and associated factors among older adults in Gedeo zone, southern Ethiopia: A mixed methods approach

Tsion Mulat Tebeje[1]*, Mihret Fikreyesus[2], Temesgen Leka Lerango[1], Daniel Sisay[1]

1 School of Public Health, College of Health Science and Medicine, Dilla University, Dilla, Ethiopia,
2 Department of Nursing, College of Health Science and Medicine, Dilla University, Dilla, Ethiopia

* yemarina12@gmail.com, Tsionmulat@du.edu.et

## Abstract

### Background

Hypertension often goes undetected for years because its initial symptoms are usually subtle and easily overlooked. Undiagnosed hypertension is a significant contributor to the onset of cardiovascular disease, renal disease, and overall mortality. Although its prevalence increases with age, few studies have investigated the factors associated with undiagnosed hypertension in older adults. Therefore, this study aimed to identify factors influencing undiagnosed hypertension among older adults in the Gedeo zone, Southern Ethiopia.

### Methods

A study design incorporating a community-based cross-sectional study and qualitative inquiry as a complementary approach was carried out among 609 randomly selected older adults in the Gedeo zone from March 19 to May 20, 2023. A binary logistic regression model assessed the relationships between the outcome and explanatory variables, with statistical significance set at a p-value of < 0.05. The qualitative data were transcribed, translated into English, and analyzed using Open Code version 4.03.

### Results

The prevalence of undiagnosed hypertension among older adults was 39.24% (95% CI: 35.43%, 43.19%). The determinants that were found to have a significant relationship with undiagnosed hypertension were living in urban areas (AOR = 0.54, 95% CI: 0.34, 0.83), being able to read and write (AOR = 0.21, 95% CI: 0.11, 0.38), attending primary education and above (AOR = 0.53, 95% CI: 0.32, 0.87), not having a health seeking behavior (AOR = 2.26, 95% CI: 1.48, 3.43), being overweight or obese

**Data availability statement:** "All relevant data are within the paper and its Supporting Information files."

**Funding:** This research was supported by a Health Professionals Education Partnership Initiative (HEPI) grant (grant number: R25TW011214) obtained from the US National Institutes of Health, Fogarty International Center. The funder had no role in study design, data collection and analysis, decision to publish, or preparation of the manuscript.

**Competing interests:** The authors have declared that no competing interests exist.

**List of abbreviations:** AOR: adjusted odds ratio; BMI: body mass index; CI: Confidence interval; DALY: disability adjusted life year; DBP: diastolic blood pressure; HTN: hypertension; IRB: institutional review board; NCD: non-communicable disease; SBP: systolic blood pressure; VIF: variance inflation factor

(AOR = 4.50, 95% CI: 2.74, 7.39), having chronic diseases (AOR = 1.72, 95% CI: 1.11, 2.66), and having a family history of hypertension (AOR = 1.90, 95% CI: 1.13, 3.21).

## Conclusion

Our findings showed that about four out of 10 older adults in the Gedeo zone have had undiagnosed hypertension. This highlights the need for a comprehensive and precise screening program for these vulnerable population. In-depth interviews revealed that television programs and healthcare providers were major sources of information regarding hypertension and its prevention methods. Therefore, it is crucial to prioritize preventive interventions and develop appropriate programs focused on older adults. In particular, people with chronic diseases and a family history of hypertension should be taught and encouraged to undergo timely checkups.

## Introduction

Non-communicable diseases (NCDs) are widely recognized as major hindrances to socioeconomic progress on a global scale [1]. In 2019, NCDs constituted seven out of the top ten leading causes of death worldwide, marking a substantial increase compared to that in the year 2000, when only four of the top ten diseases, including hypertension and diabetes, were classified as non-communicable [1,2]. Hypertension, also referred to as high blood pressure, is a chronic medical condition characterized by increased blood pressure in the arteries. This condition forces the heart to exert more effort than normal to pump blood through vessels [3].

The number of individuals aged 30–79 with hypertension grew from 648 million to 1.28 billion between 1990 and 2019, with a prevalence of 32% in women and 34% in men. Even though the treatment and control rates of hypertension have been improved, Oceania, South Asia, and sub-Saharan Africa (SSA) showed the lowest rates of detection, treatment, and control. In SSA, 52% of women and 66% of men are undiagnosed yet hypertensive [4]. The pooled prevalence of hypertension in Africa among adults aged 50 years or above was 57%, ranging from 53% in western Africa to 78% in southern Africa [5].

Hypertension is a prominent contributor to global mortality. According to 2016 Global Health Observatory data from the WHO, high blood pressure accounts for approximately 7.5 million deaths annually, accounting for approximately 12.8% of all deaths. Furthermore, this translates to 57 million disability-adjusted life years (DALYs), representing 3.7% of the total DALYs [6]. Hypertension is a common chronic disease among elderly people, and the risk increases with age, with a prevalence of 22.4%, 54.5%, and 74.5% among those aged 18–39, 40–59 and 60 and above, respectively [7].

Individuals with undiagnosed hypertension are hypertensive but have not been informed by a healthcare professional about their condition [8]. It is an issue in both developed and developing nations and contributes up to 27% of the increasing

burden of cardiovascular disease in SSA [9]. Only 10% of individuals had successfully managed to control their hypertension in low- and middle-income countries [10].

Undiagnosed hypertension significantly increases the risk of complications, including myocardial infarction, heart failure, renal failure, stroke, and premature death [11–13]. It has serious health and financial consequences; older people are disproportionately affected by high blood pressure, a recognized cardiovascular disease risk factor [14,15]. Early detection and diagnosis are therefore essential for managing hypertension; however, in poor nations such as Ethiopia, the majority of populations are ignorant of their condition, leading to undiagnosed, untreated, and unregulated hypertension [16]. A study in India showed that those aged above 50 years have a six-fold greater risk of undiagnosed hypertension [17]. According to a systematic review and meta-analysis conducted in Ethiopia, older individuals were at increased risk of undiagnosed hypertension [18].

In Ethiopia, among adults aged 30–79 years, 27.4% are hypertensive. Of which only 37.4% of women and 30.1% of men are diagnosed, only 15.6% of women and 15.6% of men are treated, and only 6.6% of women and 5.6% of men are controlled [4]. The design of hypertension interventions in Ethiopia ignores the greater population of hypertensive persons who are not diagnosed, focusing primarily on those who have been diagnosed. Given that many people with hypertension are likely to be misdiagnosed, this might have major repercussions for the nation.

Chronic diseases have a greater impact on older adults, as they are both more prevalent and more detrimental in this age group [19]. For example, as per a study in Southern Africa, chronic diseases are more frequent among adults aged 50 and beyond than among the younger population aged 18–49 [20]. NCDs, including hypertension, are responsible for the death of 41 million people globally, of these 17 million people die before age 70 [21]. The longstanding effect of hazardous health practices and current health choices increases the risk of developing a chronic illness in older age [22,23]. Yet, little is known about the prevalence and determinants of undiagnosed hypertension among adults aged 50 years and above.

Although undiagnosed hypertension is common in older adults, to our knowledge, no studies have been conducted in Ethiopia focusing on this population. Additionally, by supplementing this study with a qualitative approach, contextual factors from the perspective of elderly people can be explored. The outcomes of this study will help institutions develop appropriate hypertension disease plans and interventions. Therefore, the objective of this study was to evaluate the burden and associated factors of undiagnosed hypertension among older adults by using a mixed approach in the Gedeo zone, southern Ethiopia.

## Methods and materials

### Study area, design, and period

The study was carried out in the Gedeo Zone, which is located in the southern region of Ethiopia. 360 km from the capital of Addis Ababa. The zone encompasses 5,890.2 km² in total. There are five town administrations (Dilla town, Yirgachefe, Gedeb, Wonago, and Chelelektu) and eight districts (Wonago, Repe, Yirgacheeffe, Chorso, Bule, Kochire, Dilla zuriya, and Gedeb woredas) in the zone. With a crude population density of 774 people per square kilometer, the Gedeo zone has 1,226,779 people living in it overall, with a population of 107,312 elderly people. The Gedeo Zone Health Department reported that there were 250,363 households in the zone as of 2021. A cross-sectional study based in the community was carried out from March 19 to May 20, 2023, which was further supplemented with a qualitative study.

### Population

The source population was all older adults aged 50 years and older living in Gedeo zone, southern Ethiopia. Older adults aged 50 years and older living in Gedeo zone who were randomly selected at the time of the study period composed the study population (S1 Dataset). The study included older adults who had resided in the study area for a minimum of six months before the survey, while those with a history of hypertension and severe medical conditions or mental illness were excluded.

## Sample size and sampling procedure

The single population proportion formula was used to calculate the sample size at a confidence level of 1.96 and a margin of error of 5%. We considered 50% of the population proportion (p), as there has been no previously conducted research on the prevalence and associated factors of undiagnosed hypertension among older adults in Ethiopia. Then, by taking the 1.5 design effect into account and adding a 10% nonresponse rate, the final sample size became 633 (S1 File). For the qualitative study, the data was saturated when the number of participants reached 13.

A multistage sampling technique was utilized to select the study subjects (older adults). Among the zone's five city administrations and eight districts, two districts (Dilla Zuria and Wonago woreda) and one city administration (Dilla town) were selected randomly. Then, a total of fourteen kebeles were chosen from each designated county by the lottery method to obtain 30% of the kebeles in that county. Households from each kebele were chosen proportionately according to the number of households in each chosen kebele using lists of kebeles and the number of households from zone and district officials (Fig 1). Then, systematic random sampling was conducted to select the study households. By dividing the total number of households in a particular kebele by the necessary number of participants, the selection interval (k) was established. One house was randomly selected from each selected kebele as a starting household. The next step involved selecting households at regular intervals or every kth household as per the sampling fraction until the final household assigned to each kebele was chosen. If an eligible candidate was not found in a selected household, the sampling proceeded clockwise to the next household until an eligible person was identified. If there was more than one eligible person aged 50 and above in the selected household, one of them was randomly selected as a study participant.

## Study variables

The outcome variable, undiagnosed hypertension, was dichotomized into two groups: yes and no. The explanatory variables used in this study were residence, educational status, occupation, sex, marital status, household monthly income, family history of hypertension, age, khat chewing, physical exercise, cigarette smoking, alcohol consumption, comorbidities, obesity, knowledge, and health-seeking behavior related to hypertension.

## Measurements

First, we made sure that all participants avoided smoking cigarettes, consuming alcohol, or ingesting caffeine, and waited for 30 minutes before their blood pressure was measured. This waiting period ensured their eligibility for the blood pressure measurement, as it was verified that they had refrained from consuming anything within the last half hour. Moreover, participants were instructed to empty their bladders and abstain from drinking water before the measurement. Upon entering the measurement room, participants were asked to sit in a chair with their feet flat on the floor and their backs supported for 3 minutes before the first blood pressure reading was taken [24]. Blood pressure was measured while the participants were in a sitting position using a standard mercury sphygmomanometer. The measurement was taken from the right arm, and an appropriate cuff size was used, ensuring that it covered at least two-thirds of the upper arm. Two measurements were made 5 minutes apart. The mean systolic and diastolic blood pressures were measured in accordance with the WHO guidelines. A systolic blood pressure (SBP) measurement equal to or greater than 140 mmHg and/or a diastolic blood pressure (DBP) measurement equal to or greater than 90 mmHg were used to diagnose hypertension [25]. The participant was considered undiagnosed if the person had hypertension but did not disclose receiving a diagnosis from a medical expert.

An electronic scale or balance was used to measure weight, and a standing posture was used to assess height. Body mass index (BMI) was calculated as the weight of adults in kilograms divided by their height in meters squared. A BMI less than 18.5 kg/m2 was classified as underweight, 18.5-24.9 kg/m2 as normal, 25-29.9 kg/m2 as overweight, and 30 kg/m2 or more as obese [26].

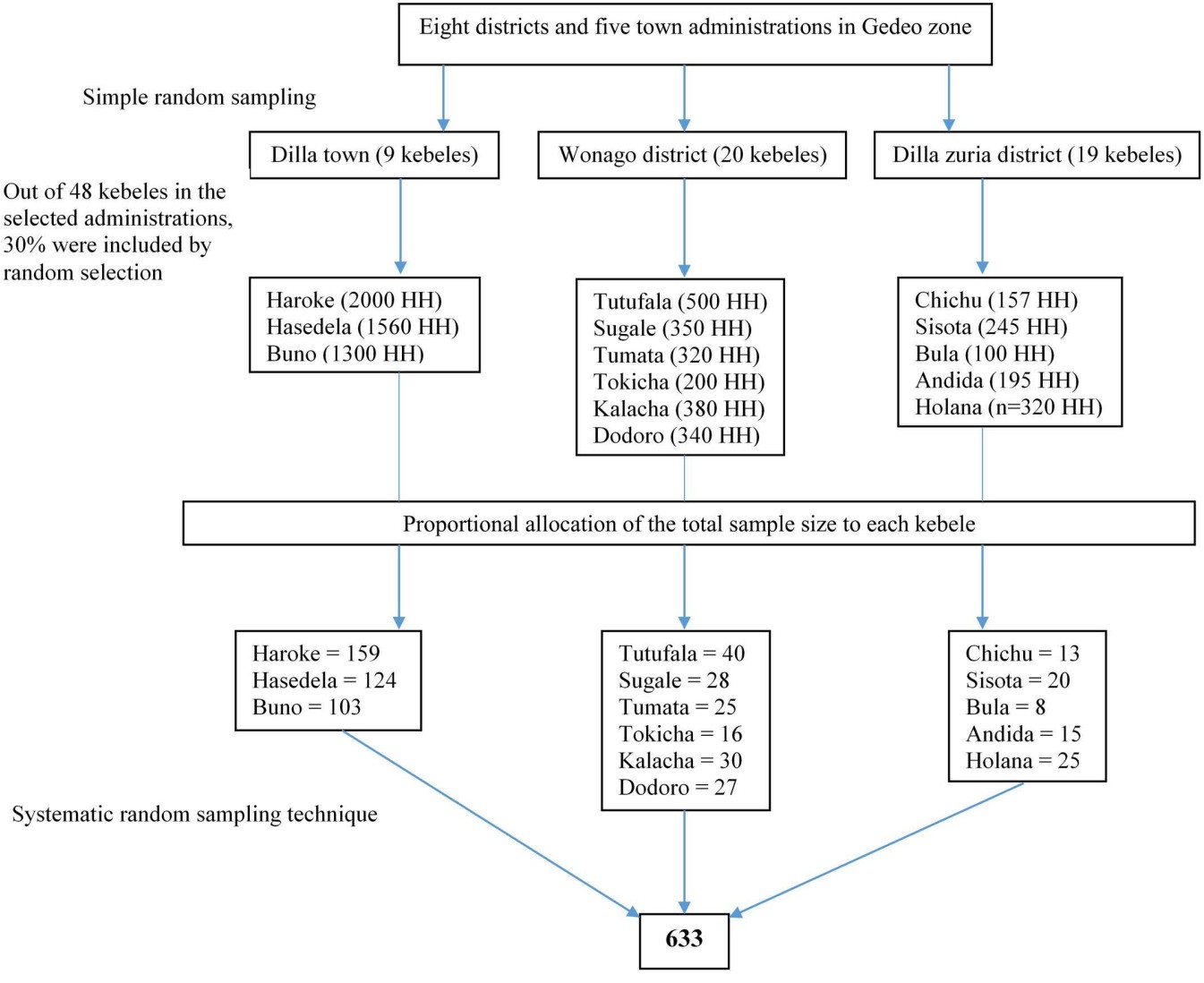

**Fig 1. Schematic representation of the sampling technique to select older adults in Gedeo zone, southern Ethiopia.**

An individual was classified as having a family history of hypertension if they had a first-degree relative diagnosed and/or medicated for the condition [27]. Comorbid disease is a chronic illness with a verified diagnosis of a condition other than hypertension [28].

Engaging in physical activity for 20–30 minutes, three days a week is considered regular physical exercise [29]. The definition of current alcohol use was any amount or kind of alcohol consumed within a year before the study. An adult who had smoked cigarettes within a year before the study was referred to as a current smoker [30]. Regular khat chewing refers to chewing khat at least once a week for the previous year or longer [31].

Health-seeking behavior was defined as follows: "yes" when they claimed to visit modern health institutions (private clinics, hospitals, and health centers), "yes" when any household member was sick, or "no" otherwise [32].

Knowledge about hypertension: Knowledge of risk factors for HTN was measured by 11 items, 5 items assessed knowledge of HTN symptoms, 3 items examined knowledge of HTN complications, and 2 items rated knowledge of HTN

medical treatment. Each right response received a score of one, while wrong and I Do Not Know responses received a score of zero. This created a range of 0–21 for the overall knowledge score. According to the guidelines provided by the instrument, a total knowledge percentage score of at least 70% (≥14.7 out of 21) denotes appropriate knowledge, whereas a score of less than 70% (≤14.7 out of 21) denotes inadequate knowledge. The internal consistency of the original questionnaire was 0.70 [33].

## Data collection tools, procedures, and quality assurance

The data collection process involved conducting face-to-face interviews using a structured questionnaire. The questionnaire was developed by drawing from various previously conducted studies and the literature. It was then adapted to suit the specific local context of the study area. The interviews aimed to gather information related to the sociodemographic status, behavioral factors, and knowledge of the study participants. It was initially written in English, and for the purpose of collecting the data, a language expert translated it into Amharic and Gedeo'ffa. The questionnaire was then translated back into English by a different language specialist to ensure uniformity.

We used KoboCollect to collect the data in this study. For data collection, 14 health extension workers and for supervision three nurses were selected. A comprehensive three-day training program was conducted to equip data collectors and supervisors with the necessary skills and knowledge regarding the overall field survey techniques, interviewing approach, measurement, steps, and process of data collection. To ensure the reliability of the data collection tools, a pretest was conducted on 5% of the total sample size in Bule Woreda 01 Kebele. This was carried out two weeks before the actual data collection period. Additionally, the principal investigator and supervisors provided daily feedback and corrections to the data collectors before they were deployed to the field the following day.

Using a semistructured interview guide, in-depth interviews (IDIs) were used to gather qualitative data. For individuals who were eligible, the purpose of the research was described. After that, the interviewers purposefully selected the participants and included them in the IDI. The population for the qualitative study was a subset of elderly individuals living in randomly selected kebeles who were not hypertensive after blood pressure measurement. Thirteen IDIs were conducted with them to understand their perceptions of the knowledge and prevention of hypertension. We obtained written informed consent from each subject. During each session of the IDIs, the conversations were recorded using a phone recorder. In addition, keynotes were taken to capture important points and details. The interviews took place in a private room that was specifically arranged to provide comfort and confidentiality for the participants.

## Data management and analysis

The data analysis was conducted using STATA version 17, a statistical software package, to process and analyze the collected data. The data were cleaned, encoded, and recoded. Appropriate descriptive statistics, such as frequency, percentage, mean, and standard deviation, were employed to examine the distribution of the data and provide a comprehensive summary. To investigate the determinants of undiagnosed hypertension, we utilized a binary logistic regression model. Initially, a bi-variable binary logistic regression analysis was conducted to identify variables that were eligible for inclusion in the multivariable analysis. Variables with a p-value less than 0.20 in the bi-variable analysis were considered potential candidates for the multivariable binary logistic regression analysis. In the multivariable analysis, variables with a p-value less than 0.05 were deemed to be statistically significant predictors. The results are presented as crude and adjusted odds ratios, accompanied by their corresponding 95% confidence intervals (CIs). Multicollinearity, which is the presence of high intercorrelations between independent variables in the multiple regression model, was assessed and quantified by the variance inflation factor (VIF). All variables demonstrated VIF values below 10, with a mean VIF of 1.91, indicating that multicollinearity was not a significant concern in the analysis.

For the qualitative data, all recorded interviews were transcribed verbatim into the local language Amharic and Gedeo'ffa. Then, the questionnaire was translated into the English language. To thoroughly understand the content, we

extensively reviewed the transcripts and created concise memos and codes for each line. Subsequently, we refined and compared the newly emerged themes and subthemes. The investigators engaged in daily discussions to resolve any discrepancies, incorporated new ideas from the participants, and reached a consensus on the identified themes and sub-themes. For the data analysis, we utilized open code version 4.03 software, employing a thematic analysis approach that followed an inductive methodology.

### Ethical consideration

This study was conducted following the principles outlined in the Declaration of Helsinki. The institutional review board (IRB) of Dilla University approved the study (protocol unique number: duirb/020/23–02). The survey questionnaire included comprehensive information on consent, confidentiality, and the survey's objectives, which were communicated on the first page. Participants were informed that their involvement was voluntary and that they had the right to withdraw their participation at any time. Prior to participation, each individual provided written informed consent. Confidentiality was strictly maintained by ensuring that the collected data did not include any personal identifiers and was solely utilized for research purposes.

## Results

### Characteristics of the study participants

We analyzed data on a total of 609 participants, for a 96.2% response rate. The mean age in the sample was $60.5 \pm 10.8$ years, while the majority (63.4%) were aged between 50 and 59 years. Almost half (49.1%) of the participants were men, and the other half (50.9%) were women. The vast majority of the participants (82.6%) were married. Of the total sample, 46.6% lived in rural areas, and 37.9% had attained primary education or above. Approximately 286 (44.0%) participants were unemployed, followed by private workers (24.0%). Nearly half of the participants (49.9%) were Gedeo in ethnicity, followed by Gurage and Oromo (Table 1).

Among the respondents, 322 (52.9%) chewed khat, 160 (26.3%) smoked cigarettes, 282 (46.3%) drank alcohol, and only one-third exercised physically. Regarding body mass index, more than two-thirds of the participants had a BMI $< 25$ kg/$m^2$. More than half (55.2%) of the participants had health-seeking behavior, and 246 (40.4) had at least one chronic disease. Among the total sample, 84 (13.8%) reported a history of hypertension in their first-degree relative (Table 1).

**Participants' knowledge of hypertension.** Of the participants, only 192 (31.5%) were aware that alcohol use increases the risk of hypertension. Nearly one-third (201, 33.0%) of the respondents knew that the chance of developing hypertension increased when first-degree relatives had the disease. Almost half of the participants (297, 48.8%) knew that excessive intake of salt could cause hypertension. Among the respondents, 217 (35.6) were aware that symptoms of HTN are not always present, and half (50.4%) recognized that blood pressure is deemed high if it is 140/90 mmHg or above, while only 14.6% identified that blood pressure 120/80 is considered good. Only 71 (11.7%) knew that when a person has two or more increased blood pressure readings on three distinct occasions, they may be diagnosed with HTN. The vast majority of 513 (84.2%) did not know that hypertension is a life-threatening condition, and only 73 (12%) HTN cases had detrimental effects on the body by causing damage to blood vessels. Among the participants, only 122 (20.4%) people were aware that using antihypertensive drugs for an extended period of time damages the body, and 261 (42.9%) were aware that there are several kinds of these medications (S1 table).

According to the instrument, a total score of hypertension knowledge below 70% (14.7) was regarded as inadequate knowledge. Overall, 539 (88.5%) of the respondents had inadequate knowledge, while the remaining 11.5% had adequate knowledge (S1 table).

**The magnitude of undiagnosed hypertension.** The study participants had mean SBP and mean DBP values of 129.8 mmHg (SD = 12.0) and 84.3 mmHg (SD = 8.2), respectively. According to this study, elderly individuals had an undiagnosed HTN incidence of 39.24% (95% CI: 35.43%, 43.19%) (Fig 2).

**Table 1. Characteristics of the study participants in the Gedeo zone, southern Ethiopia, 2023.**

| Variables | Categories | Frequency | Percent (%) |
|---|---|---|---|
| Age | 50-59 | 386 | 63.4 |
| | 60-69 | 106 | 17.4 |
| | >=70 | 117 | 19.2 |
| Sex | Female | 310 | 50.9 |
| | Male | 299 | 49.1 |
| Residence | Rural | 284 | 46.6 |
| | Urban | 325 | 53.4 |
| Marital status | Married | 503 | 82.6 |
| | Widowed | 106 | 17.4 |
| Educational status | Cannot read and write | 245 | 40.2 |
| | Can read and write | 133 | 21.9 |
| | Primary education and above | 231 | 37.9 |
| Occupation | Farmer | 101 | 16.6 |
| | Unemployed | 286 | 44.0 |
| | Private | 146 | 24.0 |
| | Others | 76 | 12.4 |
| Ethnicity | Gedeo | 304 | 49.9 |
| | Gurage | 116 | 19.1 |
| | Oromo | 102 | 16.8 |
| | Others | 87 | 14.3 |
| Khat chewing | Yes | 322 | 52.9 |
| | No | 287 | 47.1 |
| Cigarette smoking | Yes | 160 | 26.3 |
| | No | 449 | 73.7 |
| Alcohol drinking | Yes | 282 | 46.3 |
| | No | 327 | 53.7 |
| Physical exercise | Yes | 188 | 30.9 |
| | No | 421 | 69.1 |
| Health seeking behavior | Yes | 336 | 55.2 |
| | No | 273 | 44.8 |
| Overweight and obesity | Yes | 205 | 33.7 |
| | No | 404 | 66.3 |
| Chronic diseases | Yes | 246 | 40.4 |
| | No | 363 | 59.6 |
| Family history of HTN | Yes | 84 | 13.8 |
| | No | 525 | 86.2 |

## Factors associated with undiagnosed hypertension

After adjusting for the effect of confounders, the results of multivariable binary logistic regression analysis showed that educational status, place of residence, health-seeking behavior, overweight and obesity status, family history of hypertension, and the presence of at least one chronic disease were significant predictors of undiagnosed hypertension among elderly individuals. We found that elderly people who lived in urban areas (AOR = 0.54, 95% CI: 0.34, 0.83) had 44% lower odds of having undiagnosed hypertension than those who lived in rural areas. The odds of experiencing undiagnosed hypertension among elderly people who could read and write and attend primary education and above were reduced by

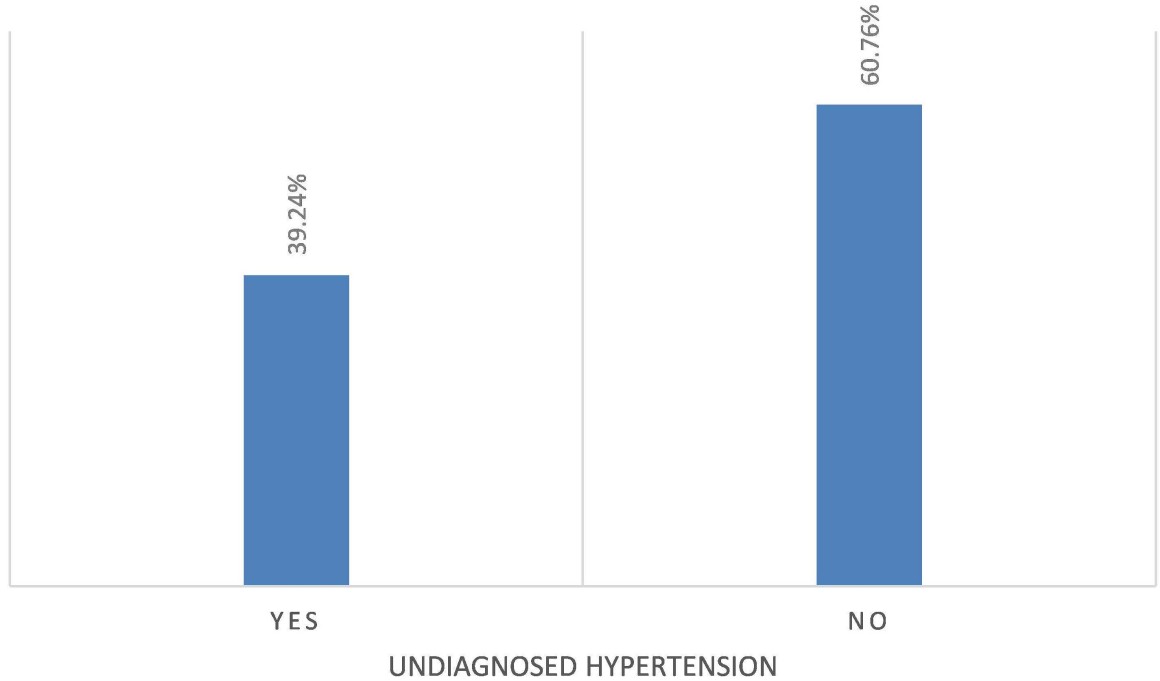

**Fig 2. Prevalence of undiagnosed hypertension among older adults in Gedeo zone, southern Ethiopia, 2023.**

79% (AOR = 0.21, 95% CI: 0.11, 0.38) and 47% (AOR = 0.53, 95% CI: 0.32, 0.87), respectively, compared to elderly people who could not read and write. In contrast to elderly individuals who exhibited health-seeking behavior, those who did not have HSB had 2.26-fold (AOR = 2.26, 95% CI: 1.48, 3.43) greater odds of undiagnosed hypertension. Compared to elderly individuals who were neither overweight nor obese, those who were overweight or obese were 4.5 times (AOR = 4.50, 95% CI: 2.74, 7.39) more likely to have undiagnosed hypertension. Additionally, the likelihood of undiagnosed hypertension was 72% and 90% greater for elderly individuals with chronic diseases (AOR = 1.72, 95% CI: 1.11, 2.66) and a family history of hypertension (AOR = 1.90, 95% CI: 1.13, 3.21), respectively, than for their counterparts (Table 2).

## Qualitative results

A key informant interview was conducted, and the data were saturated when the number of participants reached 13. Eight of them were female, while the rest were male. After reading the transcripts, two themes were developed that had different subthemes.

**Theme 1: Perception towards HTN. Subtheme 1: Definition of HTN**

Almost all of the participants were aware that hypertension occurred when blood vessel pressure was higher than usual. However, many are unaware of what a normal blood pressure level is. Only a few participants stated that hypertension occurs when the pressure in the blood vessel is more than 120/80. The majority of them learned about the definition of hypertension from watching health-related TV shows. Others heard about it from other friends, relatives, and coworkers who had the disease and from medical professionals. This is how one participant put it.

*"Regarding my blood pressure situation…. I know the normal reading for blood pressure is 120/80, and it should not be more than that... If it is greater than this, I understand it is not a good thing. In addition, I always measure my blood pressure…I heard this from a TV show at one time." (Participant 1, Male age 58)*

**Table 2. Multivariable binary logistic regression analysis of factors associated with undiagnosed hypertension among elderly individuals in the Gedeo zone, 2023.**

| Variables | Category | Undiagnosed hypertension Number (%) | COR (95%CI) | AOR (95%CI) |
|---|---|---|---|---|
| Sex | Female | 130 (54.4) | 1 | 1 |
| | Male | 109 (45.6) | 0.74 (0.57, 1.10) | 1.53 (0.99, 2.34) |
| Residence | Rural | 119 (49.8) | 1 | 1 |
| | Urban | 120 (50.2) | 0.81 (0.59, 1.12) | **0.54 (0.34, 0.83)**** |
| Educational status | cannot read and write | 112 (46.9) | 1 | 1 |
| | can read and write | 39 (16.3) | 0.49 (0.31, 0.77) | **0.21 (0.11, 0.38)***** |
| | primary education and above | 88 (36.8) | 0.73 (0.51, 1.05) | **0.53 (0.32, 0.87)*** |
| Occupation | Farmer | 33 (13.8) | 1 | 1 |
| | Unemployed | 139 (58.2) | 1.95 (1.21, 3.14) | 1.57 (0.92, 2.67) |
| | Private | 39 (16.3) | 0.75 (0.43, 1.31) | 0.57 (0.31, 1.06) |
| | Others | 28 (11.7) | 1.20 (0.64, 2.24) | 1.38 (0.66, 2.87) |
| Health seeking behavior | Yes | 109 (45.6) | 1 | 1 |
| | No | 130 (54.4) | 1.89 (1.36, 2.63) | **2.26 (1.48, 3.43)***** |
| Overweight and Obesity | No | 53 (22.2) | 1 | 1 |
| | Yes | 186 (77.8) | 2.45 (1.69, 3.54) | **4.50 (2.74, 7.39)***** |
| Chronic disease | No | 112 (46.9) | 1 | 1 |
| | Yes | 127 (53.1) | 2.39 (1.71, 3.34) | **1.72 (1.11, 2.66)*** |
| Family history | No | 193 (80.8) | 1 | 1 |
| | Yes | 46 (19.3) | 2.08 (1.31, 3.31) | **1.90 (1.13, 3.21)*** |

*** < 0.001,

** < 0.01 and

* < 0.05, COR crude odds ratio, AOR adjusted odds ratio

## Subtheme 2: Symptoms and fear of HTN

The majority of participants listed headache and dizziness as common symptoms of hypertension. In addition, they were only able to state the aforementioned symptoms. Fainting and blurred vision were mentioned as signs of high blood pressure by few of the participants. The majority of participants stated that they would be anxious to learn that they had the illness since it could lead to serious complications and require them to modify their diet and activities. They become increasingly anxious about being diagnosed with this disease, as they see the illness as causing more health issues and even death in other people. A few mentioned that it might happen to them, and since it can happen to anyone, it is nothing to be alarmed about; all they need to do is take their medication and adjust their way of living. One of the participants expressed that she would be very stressed if she were to be diagnosed with hypertension as follows:

> *"I lost my father ten years ago due to hypertension. I truly hate that disease…he was just fine the whole day and he got angry at night because of something and he fainted…after that, he was admitted to hospital for 10 days...the doctors said he had a stroke….he could not get out of the hospital alive…so I don't want to be diagnosed with the disease." (Participant 10, female 52 years)*

Another participant described it as follows: *"Hypertension is a silent killer. I heard from my friend, families… I don't know what it may bring on me. So I take care of myself as much as possible." (Participant 5, male 70 years)*

**Theme 2: Prevention and management of HTN.  Subtheme 1: Preventive activities**

The majority of participants indicated that by engaging in various preventive activities, they could avoid hypertension. Most of them stated that as they get older, all they need to do is reduce their salt intake, stress, and exercise. Few participants claimed that cutting back on alcohol, coffee, and fatty meats would help them avoid the illness. One participant said that to avoid the disease, he must maintain his weight within a normal range for his height and age. One of the interviewees put it this way:

*"I don't know if it is scientific or not. However, I know there are three things that are said to be deadly and lead to increased blood pressure… the three white…Like salt, sugar, and fatty meat… I will control myself from these things"* *(Participant 6, male 65 years).*

**Subtheme 2: Self-management**

The participants were asked what they would do if they were to be diagnosed with hypertension. They all said they needed to exercise, reduce salt, and follow the advice of health care providers. Half of the participants added that patients must adhere to health care provider's orders and take their medication as prescribed strictly. Very few participants stated that they needed to reduce their alcohol and coffee intake, stop smoking cigarettes, and start checking their blood pressure more frequently. One participant expressed it as follows:

*"It is like what I said before. By doing physical exercise and modifying one's feeding style…avoiding different stressful things...taking things easily…people can manage high blood pressure"* *(participant 12, male 70 years old).*

## Discussion

Older adults are at increased risk of hypertension. As age increases, the vascular system undergoes changes that result in stiffening of the arteries, leading to a rise in blood pressure [34]. The purpose of this study was to determine the prevalence of and determinants of undiagnosed hypertension among elderly people by conducting qualitative and quantitative cross-sectional studies. According to this study, 39.24% (95% CI: 35.43%, 43.19%) of the older adults had undiagnosed hypertension. In other words, close to forty percent of older adults included in this study were hypertensive but did not know about their disease. This finding is greater than those of studies performed among elderly people in Iran [35] and China [36]. The disparity could be attributed to socioeconomic differences between countries. Compared with those countries, Ethiopia has lower education, healthcare system, infrastructure, and access to medical services, which increases the magnitude of undiagnosed hypertension [37]. However, this value is lower than that reported in a study from Bangladesh [38]. This discrepancy could be explained by the difference in the scope of the study and population, as their participants were aged 35 + years.

The results of the multivariable binary logistic regression model showed that elderly individuals who lived in urban areas had a lower likelihood of having undiagnosed hypertension than those who lived in rural areas. This finding aligns with previous studies performed in India [39], China [36] and Malaysia [40]. This is because urban regions have more access to healthcare facilities than rural regions. Additionally, the inaccessibility of healthcare facilities as a result of poor transport and communication as well as the poor use of healthcare services in rural regions due to reliance on traditional medicines contributes to undiagnosed hypertension [41,42].

Consistent with previous studies [40,43,44], elderly people who can read and write and/or attend primary education have greater chances of having undiagnosed hypertension than elderly people who are unable to read and write. This shows that a low educational level is associated with a lack of awareness and limited understanding of health-related information. Few of the participants knew the scientific and correct definition of high blood pressure. They simply understand that it is an increase in blood pressure. Few participants stated that hypertension occurs when their blood pressure is greater than 120/80. They

learned this from television shows, health care providers, and others around them, which is similar to the findings of a study performed in Uganda [45]. However, in Malaysia, most of the participants acquired their knowledge about hypertension from the Internet [46]. All participants identified headaches and dizziness as the most common symptoms of hypertension, with only a few participants mentioning additional symptoms such as blurred vision and fainting, similar to the findings of a study performed in Malaysia [46]. However, none of them were aware of additional symptoms such as chest pain, difficulty breathing, vomiting, nausea, or other symptoms. The majority of participants are afraid of acquiring this disease because it causes additional complications, sudden death, and lifestyle changes, similar to a study performed in Uganda [45].

It was also found that compared to elderly individuals who exhibit health-seeking behavior, those with no health-seeking behavior are at high risk of having their hypertension undiagnosed. This is corroborated by studies from Ethiopia [11,47]. The possible reason is that those who have low healthcare-seeking behavior are less likely to visit health institutions to undergo hypertension screening.

Our findings support those of studies conducted in Ethiopia [47–50], which reported higher odds of undiagnosed hypertension among those who were overweight or obese. Excess body weight is closely linked to an increased risk of cardiovascular morbidity and an elevated cardiovascular risk. There is a well-established association between being overweight and the activation of the sympathetic nervous system and the renin-angiotensin system, leading to the development of hypertension. Neuroendocrine pathways are the main way that obesity and hypertension are related, and recent research has suggested that factors originating from adipose tissue play a significant role in this relationship [51,52].

Chronic disease was also found to have a significant influence on undiagnosed hypertension. In this regard, elderly individuals with any chronic disease had increased odds of undiagnosed hypertension. This finding was consistent with the results of previous studies [18,27,47,49]. For instance, among those with diabetes mellitus, elevated blood glucose levels can indeed cause extensive damage to blood vessels, increase fluid volume, and promote insulin resistance, all of which are key mechanisms contributing to the increased risk of hypertension [53].

Furthermore, the likelihood of undiagnosed hypertension was greater among elderly individuals with a family history of hypertension. This was in agreement with studies from Ethiopia [18,27,48], and a possible explanation might be that the greater chance of undiagnosed hypertension among those who have a family history of HTN may be linked to genetic factors that increase the risk of developing high blood pressure. The underlying genetic mechanisms are likely to involve variations in multiple genes that impact the regulation of blood pressure and cardiovascular system function [54].

The majority of participants were aware of disease prevention measures. They stated that they should exercise more, consume less salt, and avoid stress. Others stated that they needed to reduce their consumption of alcohol and cigarettes. One participant stated that maintaining a normal weight-to-height ratio is one way to prevent the disease. This finding is in line with research performed in Aksum, Ethiopia [55]. Likewise, the majority of participants stated that if they were diagnosed with hypertension, they would simply need to adjust their lifestyle based on the advice of their doctors and strictly adhere to their medication. Others stated that physical activity will help manage the disease, which is comparable to the findings of a study carried out in Uganda [45].

## Strengths and limitations

The study was not without limitations. Due to its cross-sectional design, we were unable to establish cause-and-effect relationships. The notable strengths were the community-based approach, utilization of mixed study, and relatively large sample size. Additionally, standard guidelines were followed for diagnosing hypertension and assessing the other variables.

## Conclusion

This study revealed that about four out of 10 older adults have had undiagnosed hypertension in Gedeo zone of Ethiopia. This showed the importance of implementing a comprehensive and precise screening program for these susceptible population. Place of residence, educational status, health-seeking behavior, overweight and obesity status, family history of

hypertension, and presence of at least one chronic disease were significantly associated with undiagnosed hypertension among elderly people. Thus, given the rising number of elderly people in low-income countries like Ethiopia, it is crucial to prioritize preventive measures and creating suitable programs focused on the elderly. In particular, people with chronic diseases and a family history of hypertension need to be taught and encouraged to undergo timely checkups. It is also imperative to develop public awareness campaigns on hypertension through media channels and platforms, including radio and television programs. Furthermore, making healthy lifestyle choices should also be carefully considered, especially in the later stages of life.

## Supporting information

**S1 File. Sample size calculation.**
(DOCX)

**S1 Table. Knowledge of the study participants about hypertension.**
(DOCX)

**S1 Dataset. Primary data collected for the study.**
(XLSX)

## Acknowledgments

We would like to acknowledge the data collectors and study participants.

## Author contributions

**Conceptualization:** Tsion Mulat Tebeje, Daniel Sisay.

**Formal analysis:** Tsion Mulat Tebeje, Mihret Fikreyesus, Temesgen Leka Lerango, Daniel Sisay.

**Funding acquisition:** Tsion Mulat Tebeje.

**Writing – original draft:** Tsion Mulat Tebeje, Daniel Sisay.

**Writing – review & editing:** Tsion Mulat Tebeje, Mihret Fikreyesus, Temesgen Leka Lerango, Daniel Sisay.

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
