## [Decision Letter · Decision Letter 0]

5 Sep 2024

PONE-D-24-27240The hidden burden of hypertension-undiagnosed hypertension and associated factors among elderly individuals in Gedeo zone: A mixed methods approachPLOS ONE

Dear Dr. Tebeje,

Thank you for submitting your manuscript to PLOS ONE. After careful consideration, we feel that it has merit but does not fully meet PLOS ONE’s publication criteria as it currently stands. Therefore, we invite you to submit a revised version of the manuscript that addresses the points raised during the review process.

We look forward to receiving your revised manuscript.

Kind regards,

Amanuel Yoseph, MPH

Academic Editor

PLOS ONE

“This research was supported by a Health Professionals Education Partnership Initiative (HEPI) grant (grant number: R25TW011214) obtained from the US National Institutes of Health, Fogarty International Center.”

“We would like to acknowledge the HEPI initiative for providing funds to conduct this study.”

“This research was supported by a Health Professionals Education Partnership Initiative (HEPI) grant (grant number: R25TW011214) obtained from the US National Institutes of Health, Fogarty International Center.”

Additional Editor Comments:

I critically reviewed your article entitled “The hidden burden of hypertension-undiagnosed hypertension and associated factors among elderly individuals in Gedeo zone: A mixed methods approach” which has the potential to add to the existing body of scientific knowledge, particularly in developing countries. However, there are some limitations in your article that need addressing before publication.

1. There are several grammatical and typological errors that authors need to carefully review.

2. Authors should extensively format manuscripts based on PLOS ONE journal style, including file naming. Avoid unnecessary italicizing and capitalization throughout the manuscript.

3. Make sure that your reference contains all the necessary details and PLOS ONE style.

Decision: Major revision

Reviewers' comments:

Reviewer's Responses to Questions

**Comments to the Author**

1. Is the manuscript technically sound, and do the data support the conclusions?

Reviewer #1: Yes

Reviewer #2: Partly

Reviewer #3: Partly

2. Has the statistical analysis been performed appropriately and rigorously? 

Reviewer #1: Yes

Reviewer #2: Yes

Reviewer #3: Yes

3. Have the authors made all data underlying the findings in their manuscript fully available?

Reviewer #1: Yes

Reviewer #2: Yes

Reviewer #3: Yes

4. Is the manuscript presented in an intelligible fashion and written in standard English?

Reviewer #1: Yes

Reviewer #2: Yes

Reviewer #3: Yes

5. Review Comments to the Author

Reviewer #1: Thank you the authors working on the so called silent killer of undiagnosed hypertension.This is a prospective study so that what was the difficulty of getting consent from each indiviadual or what does it mean for publication consent is not applicable?....exclusion criteria?,operational definition? some editting of language/expert involvement is better to be involved

Reviewer #2: The study is good for the body of knowledge and abstract and introduction was written in a clear , concise and intelligible manner.

• The title should be modified to be clear

. what is your background to say elderly 50 years or above? there is other age classification for elderly population

. Does the the district (3) selected for the study is representative for the Gedeo zone (13)?

. Data management and analysis should be concise, clear and precise Eg. VIF and multicollinearity.

. For qualitative study what types of analysis method used? what does it mean thematic content analysis, it vague and not clear, please specify it. it creates mislead to qualitative result interpretation. in addition the qualitative result interpretation should be consistent/matched with analysis method.

. what the qualitative study result support? please put the qualitative result in appropriate place

. Please check the study for grammar and plagiarized content?

.Majority (63.4%) of the study participants were age between 50 to 59 years, this classified as middle age adult. does it can say elder?

. what is the sample size for second objective?

. undiagnosed HTN is new so it is possible to say prevalence of undiagnosed HTN?

Why did you merge overweight and obesity? The degree of being obese and being overweight is different and could have different impacts.

• Study period: replace the phrase “between March 19 and May 20, 2023” with “from March 19 to May 20, 2023”

• Regarding measurement of the dependent variable, two measurements were taken 5 minutes apart. I don’t think 5 minutes are enough to diagnose a patient with hypertension. Whenever possible, the diagnosis should not be made on a single office visit.

• Before measuring the blood pressure, have you addressed the protocols to measure blood pressure? For example, how do you assesses the participants to avoid smoking, caffeine and exercise for 30 min; empty bladder; remain seated and relaxed for 3–5 min? Nothing is written in this case as it could have impact on the result. Additionally, was the room Quiet with comfortable temperature?

• Regular physical activity was defined as Moderate intensity aerobic exercise (walking, jogging, cycling, yoga, or swimming) for 30 minutes on 5–7 days per week but in your case it is “Engaging in physical activity for 20–30 minutes, three days a week, is considered regular physical exercise”

Reviewer #3: Although the manuscript is good, I have several criticisms. thus you'll attempt to review and edit in light of the feedback.

I made to attach all comments in PDF format. concept, data analysis, and following the rules when writing the manuscript are some of those.

6. PLOS authors have the option to publish the peer review history of their article (what does this mean? ). If published, this will include your full peer review and any attached files.

**Do you want your identity to be public for this peer review?** For information about this choice, including consent withdrawal, please see our Privacy Policy .

Reviewer #1: No

Reviewer #2: No

Reviewer #3: **Yes: ** Wasihun Kindalem Work

---

## [Author Response · Author response to Decision Letter 1]

28 Oct 2024

Subject: Responses to comments

Dear Editor,

Thank you for taking the time to consider our manuscript titled “Undiagnosed hypertension and associated factors among older adults in Gedeo zone, southern Ethiopia: A mixed methods approach” for the Plos One Journal original research article. We appreciate the time and effort you and the reviewers have dedicated to providing valuable feedback on our manuscript.

We have considered the comments and concerns and made every effort to address them. We agree with all the comments and have incorporated the corresponding revisions into the revised manuscript. All revised texts are track-changed to point out the changes we made. We believe that our manuscript has been significantly improved as a result of these revisions.

We would like to thank you once again for your consideration of our work and for inviting us to submit the revised manuscript. We look forward to hearing from you. Our detailed, point-by-point responses to the comments are given below.

Best regards,

Tsion Mulat Tebeje

School of Public Health, Dilla University, Dilla, Ethiopia.

Email: yemarina12@gmail.com / Tsionmulat@du.edu.et (corresponding author)

Response to journal’s requirements

Response: We prepared the manuscript according to the journal's requirements and double-checked its compliance before submitting our revised manuscript.

“This research was supported by a Health Professionals Education Partnership Initiative (HEPI) grant (grant number: R25TW011214) obtained from the US National Institutes of Health, Fogarty International Center.”

Response: Thank you for asking about the role of the funder. We have included the role of funder statement in the cover letter.

“We would like to acknowledge the HEPI initiative for providing funds to conduct this study.”

“This research was supported by a Health Professionals Education Partnership Initiative (HEPI) grant (grant number: R25TW011214) obtained from the US National Institutes of Health, Fogarty International Center.”

Response: We removed any funding-related text from the revised manuscript. We included the funding statement and role of funder statement in the cover letter.

4. Please include captions for your Supporting Information files at the end of your manuscript, and update any in-text citations to match accordingly.

Response: Thank you. We included a caption of the supplementary files at the end of the revised manuscript.

Edits requested on the submission

1. Thank you for uploading your study's underlying data set in the Supporting Information file "dataset.xlsx". We noticed that this file may contain potentially identifying participant information.

Before we proceed, we kindly ask that you please ensure that the data shared are in accordance with participant consent and all applicable local laws. If any data shared is not in accordance with participant consent, please remove these data and re-upload a fully anonymized data set. Please also note that spreadsheet columns with identifying information must be removed and not hidden as all hidden columns will appear in the published file.

Response: Thank you for the suggestion. The shared dataset does not contain any potentially identifying participant information such as names, addresses, images, or other unique identifiers. The dataset is fully anonymized. Additionally, the dataset aligns with participants’ consent agreements.

Additional Editor Comments:

I critically reviewed your article entitled “The hidden burden of hypertension-undiagnosed hypertension and associated factors among elderly individuals in Gedeo zone: A mixed methods approach” which has the potential to add to the existing body of scientific knowledge, particularly in developing countries. However, there are some limitations in your article that need addressing before publication.

Response: Dear editor, Thank you for giving us your valuable time and for sharing your valuable input.

1. There are several grammatical and typological errors that authors need to carefully review.

Response: Thank you. The revised manuscript has been meticulously edited and proofread to rectify quality issues, such as misspellings, grammatical errors, and unclear sentences.

2. Authors should extensively format manuscripts based on PLOS ONE journal style, including file naming. Avoid unnecessary italicizing and capitalization throughout the manuscript.

Response: Thank you for your suggestion. We prepared the revised manuscript according to the PLOS ONE requirements

3. Make sure that your reference contains all the necessary details and PLOS ONE style.

Response: Thank you for your suggestion. We have made sure the references fulfill the PLOS ONE guideline in the revised manuscript.

Response to reviewer’s comments

Comments from Reviewer 1

1. Thank you the authors working on the so called silent killer of undiagnosed hypertension.

Response: Thank you for giving us your valuable time to review our paper and for your comments and suggestions, which we got as valuable input to improve the manuscript.

2. This is a prospective study so that what was the difficulty of getting consent from each indiviadual or what does it mean for publication consent is not applicable?....

Response: Thank you. We did not face any difficulty obtaining consent, we got written consent from our participants as we explained in the ethical consideration section. We mentioned publication consent is not applicable because there was no personal identifier data or imaging that required publication consent.

3. exclusion criteria?,operational definition?

Response: Thank you for pointing this out. The inclusion and exclusion criteria were incorporated in the population section of the methods. Highlighted in Page 6, Line 113-116. We already included operational definition as “measurements” Page 8-10, Line 146-185.

4. some editting of language/expert involvement is better to be involved

Response: Thank you. We have edited the revised manuscript to rectify misspellings, grammatical errors, and unclear sentences.

Comments from Reviewer 2

1. The study is good for the body of knowledge and abstract and introduction was written in a clear , concise and intelligible manner.

Response: Thank you for giving us your valuable time to review our paper and for your comments and suggestions, which we got as valuable input to improve the manuscript.

2. The title should be modified to be clear

Response: Thank you for the suggestion. We modified the title to “Undiagnosed hypertension and associated factors among older adults in Gedeo zone, southern Ethiopia: A mixed methods approach”.

3. what is your background to say elderly 50 years or above? there is other age classification for elderly population

Response: Thank you for the insightful inquiry. Including hypertension, chronic diseases are more common among older adults (adults aged 50 and above). As proved by previous studies, the incidence of hypertension and other chronic illnesses increases as individuals age. Additionally, many studies in Africa used the age of 50 and above to define older adults. We have clearly explained the rationale for using these populations in the introduction section of the revised manuscript, page 5, lines 83-90. This study aims to measure undiagnosed hypertension among older adults who have been overlooked in previous studies. To ensure consistency, the term "elderly population" in the title was modified to "older adults".

4. Does the the district (3) selected for the study is representative for the Gedeo zone (13)?

Response: Thank you. As per the information we obtained from Gedeo zone administration, all the mentioned districts and town administrations belong to the Gedeo zone, which makes them representative of the zone.

5. Data management and analysis should be concise, clear and precise Eg. VIF and multicollinearity.

Response: Thank you. We rewrote the multicollinearity and VIF more clearly in the revised manuscript. Page 11-12, Lines 226-229.

6. For qualitative study what types of analysis method used? what does it mean thematic content analysis, it vague and not clear, please specify it. it creates mislead to qualitative result interpretation. in addition the qualitative result interpretation should be consistent/matched with analysis method.

Response: Thank you for the insightful inquiry. We would like to clarify that we employed thematic analysis for our qualitative study. The reference to thematic content analysis was made in error, and we have since corrected it to accurately reflect our use of thematic analysis. Regarding the consistency of our results with the method of analysis, we have revisited our interpretations and cross-verified them against our thematic analysis results. Each interpretation has been linked to specific themes in the revised manuscript, ensuring that the conclusions are grounded in the evidence presented.

7. what the qualitative study result support? please put the qualitative result in appropriate place

Response: Thank you for the suggestion. The results of qualitative analysis were placed after the quantitative result, page 18-21, Lines 308-362.

8. Please check the study for grammar and plagiarized content?

Response: Thank you. We have edited the revised manuscript to rectify misspellings, grammatical errors, and unclear sentences.

9. Majority (63.4%) of the study participants were age between 50 to 59 years, this classified as middle age adult. does it can say elder?

Response: Thank you for the insightful inquiry. Adults aged 50 and above are indeed not elderly rather they are adults who are relatively older (older adults). As we responded to question number 3, we explained why 50+ adults are our population, in the introduction section of the revised manuscript, page 5, lines 83-90.

10. What is the sample size for second objective?

Response: Thank you for your question. We have included the sample size calculation for both objectives in supplementary file 1 of the revised manuscript. We found that the sample size calculated for the second objective was lower, so we used the larger sample size from the first objective.

11. undiagnosed HTN is new so it is possible to say prevalence of undiagnosed HTN?

Response: Thank you for the insightful inquiry. We collect the data cross-sectionally, where the dependent and independent variables were collected simultaneously. Therefore, we could not determine whether the undiagnosed HTN was new or not. In order to calculate the incidence we were supposed to follow a group of people who were initially free from the undiagnosed HTN.

12. Why did you merge overweight and obesity? The degree of being obese and being overweight is different and could have different impacts.

Response: Thank you for the insightful inquiry. We merged being obese and overweight for the health and statistical context. In the health research context, both overweight and obesity are linked to increased risk of diseases, more specifically chronic diseases, though the degree of risk varies. This provides valuable insights for effective decision-making. Regarding the statistical perspective, merging variables leads to meeting the chi-square test criteria and improved statistical power to make comparisons.

13. Study period: replace the phrase “between March 19 and May 20, 2023” with “from March 19 to May 20, 2023”

Response: Thank you for the insightful suggestion. As per your suggestion, we modified the study period to ‘from March 19 to May 20, 2023’ in the abstract and methods section of the revised manuscript.

14. Regarding measurement of the dependent variable, two measurements were taken 5 minutes apart. I don’t think 5 minutes are enough to diagnose a patient with hypertension. Whenever possible, the diagnosis should not be made on a single office visit.

Response: Thank you for the insightful inquiry. We made the diagnosis in a single office visit to save time and avoid missing participants during follow-up visits to measure blood pressure. The repeated measurements were separated by 5 minutes. For proper measurement of blood pressure in an office in a sitting position, repeated measurements can be separated by 1-2 min (10.1161/HYP.0000000000000087). Therefore, a 5-minute interval was quite enough.

15. Before measuring the blood pressure, have you addressed the protocols to measure blood pressure? For example, how do you assesses the participants to avoid smoking, caffeine and exercise for 30 min; empty bladder; remain seated and relaxed for 3–5 min? Nothing is written in this case as it could have impact on the result. Additionally, was the room Quiet with comfortable temperature?

Response: Thank you for pointing this out. We completely agree with you. We addressed all protocols to measure blood pressure before taking the blood pressure measurement. We waited for 30 minutes, after making sure that the participants were refrained from smoking cigarettes, drinking alcohol, or taking caffeine before their BP measurement was taken. This by default made them eligible for BP measurement as we confirmed they had not taken anything within 30 minutes. The participants were also told to empty their bladder and avoid drinking water shortly before the measurement. After they entered the measurement room, the participants sat in a chair with feet flat on the floor and back supported for 3 minutes before taking the first BP reading. The second measurements were taken 5 minutes later. We included this procedure in the revised manuscript. Page 8, lines 147-154

16. Regular physical activity was defined as Moderate intensity aerobic exercise (walking, jogging, cycling, yoga, or swimming) for 30 minutes on 5–7 days per week but in your case it is “Engaging in physical activity for 20–30 minutes, three days a week, is considered regular physical exercise”

Response: Thank you for the insightful inquiry. We measured physical activity according to the International Physical Activity Questionnaire (IPAQ), which is a widely used tool designed to assess physical activity levels in various populations globally. As per the IPAQ, a minimum of at least three activities in a week for 20-30 minutes per day is needed to be considered as a physical activity.

Comments from Reviewer 3

Although the manuscript is good, I have several criticisms. thus you'll attempt to review and edit in light of the feedback. I made to attach all comments in PDF format. concept, data analysis, and following the rules when writing the manuscript are some of those.

Response: Thank you for giving us your valuable time to review our paper and for your comments and suggestions, which we got as valuable input to improve the manuscript.

1. From the title line 1, there is a reputation for “hypertension-undiagnosed hypertension”, please rewrite again to get more credit or approval?

Response: Thank you for the suggestion. We modified the title to “Undiagnosed hypertension and associated

---

## [Decision Letter · Decision Letter 1]

29 Nov 2024

PONE-D-24-27240R1Undiagnosed hypertension and associated factors among older adults in Gedeo zone, southern Ethiopia: A mixed methods approachPLOS ONE

Dear Dr. Tebeje,

Thank you for submitting your manuscript to PLOS ONE. After careful consideration, we feel that it has merit but does not fully meet PLOS ONE’s publication criteria as it currently stands. Therefore, we invite you to submit a revised version of the manuscript that addresses the points raised during the review process.

We look forward to receiving your revised manuscript.

Kind regards,

Amanuel Yoseph, MPH

Academic Editor

PLOS ONE

Journal Requirements:

Reviewers' comments:

Reviewer's Responses to Questions

**Comments to the Author**

1. If the authors have adequately addressed your comments raised in a previous round of review and you feel that this manuscript is now acceptable for publication, you may indicate that here to bypass the “Comments to the Author” section, enter your conflict of interest statement in the “Confidential to Editor” section, and submit your "Accept" recommendation.

Reviewer #2: (No Response)

Reviewer #3: All comments have been addressed

2. Is the manuscript technically sound, and do the data support the conclusions?

Reviewer #2: Yes

Reviewer #3: Yes

3. Has the statistical analysis been performed appropriately and rigorously? 

Reviewer #2: Yes

Reviewer #3: Yes

4. Have the authors made all data underlying the findings in their manuscript fully available?

Reviewer #2: Yes

Reviewer #3: Yes

5. Is the manuscript presented in an intelligible fashion and written in standard English?

Reviewer #2: Yes

Reviewer #3: Yes

6. Review Comments to the Author

Reviewer #2: Thank you very much to get the opportunity to review this important public health problem. The authors are fully addressed the comment I raised previously. but I have some suggestions to improve quality and scientific back ground of the manuscript.

1.What adds this study from qualitative finding?

2.In discussion parts the qualitative study have not support the quantitative finding simply put in separate manner and compares with other similar study only. what is the reason to add qualitative study to this research? I don.t see its relevance as it stands.

3. the objective is the same for both quantitative and qualitative, but it seems like different in your write up. it needs arrangement. for example you should explained the quantitative study, then followed by qualitative finding to make the finding strong.

4. non of the participants knew the scientific back ground of the HTN. It needs modification.

5.the qualitative analysis in some sentences seems like to summative content analysis. it needs some modification.

7.quantitative and qualitative study lack integrity at discussion parts of the manuscript. it needs align together.

Reviewer #3: (No Response)

7. PLOS authors have the option to publish the peer review history of their article (what does this mean? ). If published, this will include your full peer review and any attached files.

**Do you want your identity to be public for this peer review?** For information about this choice, including consent withdrawal, please see our Privacy Policy .

Reviewer #2: **Yes: ** Asmamaw Deguale Worku

Reviewer #3: **Yes: ** wasihun kindalem worku

---

## [Author Response · Author response to Decision Letter 2]

13 Jan 2025

Subject: Responses to comments (revision 2)

Dear Editor,

Thank you for taking the time to consider our manuscript titled “Undiagnosed hypertension and associated factors among older adults in Gedeo zone, southern Ethiopia: A mixed methods approach” for the Plos One Journal original research article. We appreciate the time and effort you and the reviewers have dedicated to providing valuable feedback on our manuscript.

We have considered the comments and concerns and made every effort to address them. We agree with all the comments and have incorporated the corresponding revisions into the revised manuscript. All revised texts are track-changed to point out the changes we made. We believe that our manuscript has been significantly improved as a result of these revisions.

We would like to thank you once again for your consideration of our work and for inviting us to submit the revised manuscript. We look forward to hearing from you. Our detailed, point-by-point responses to the comments are given below.

Best regards,

Tsion Mulat Tebeje

School of Public Health, Dilla University, Dilla, Ethiopia.

Email: yemarina12@gmail.com / Tsionmulat@du.edu.et (corresponding author)

Journal Requirements

Response: We have reviewed our list of references and changed some references. We have removed and replaced the retracted articles with relevant and current references. All changes made to the reference list have been highlighted in the revised manuscript using track change.

Response to reviewers

Reviewer 2

General comment: Thank you very much to get the opportunity to review this important public health problem. The authors are fully addressed the comment I raised previously. but I have some suggestions to improve quality and scientific back ground of the manuscript.

Response: Thank you for your kind words and for reviewing our manuscript which benefited from your comments and suggestions.

Comment 1: What adds this study from qualitative finding?

Response 1: The qualitative study aimed to assess participants' knowledge and attitudes towards hypertension. Participants' knowledge is one aspect measured in the quantitative portion of this research. Consequently, the qualitative findings provided further insight into their understanding of hypertension, including its definition, signs and symptoms, self-management strategies, and preventive measures. This approach revealed additional concepts regarding knowledge about hypertension that are not addressed in the quantitative questions.

Comment 2: In discussion parts the qualitative study have not support the quantitative finding simply put in separate manner and compares with other similar study only. what is the reason to add qualitative study to this research? I don.t see its relevance as it stands.

Response 2: Thank you. As we mentioned above the qualitative study is meant to assess participants' knowledge and attitudes towards hypertension. We aligned the qualitative and quantitative study in the discussion section.

Comment 3: the objective is the same for both quantitative and qualitative, but it seems like different in your write up. it needs arrangement. for example you should explained the quantitative study, then followed by qualitative finding to make the finding strong.

Response 3: Thank you for your insightful suggestion. In the results section, we started with quantitative results, followed by qualitative findings.

Comment 4: non of the participants knew the scientific back ground of the HTN. It needs modification.

Response 4: Thank you. We have changed it to “few of them knew the scientific definition of HTN” and we made the changes with track change to spot the changes.

Comment 5: The qualitative analysis in some sentences seems like to summative content analysis. it needs some modification.

Response 5: Thank you. As per your suggestion, we have modified the parts that may seem summative content analysis. Page, lines

Comment 6: quantitative and qualitative study lack integrity at discussion parts of the manuscript. it needs align together.

Response 6: Thank you for pointing this out. In the discussion section, we integrated the qualitative findings with the quantitative findings.

---

## [Decision Letter · Decision Letter 2]

26 Mar 2025

Undiagnosed hypertension and associated factors among older adults in Gedeo zone, southern Ethiopia: A mixed methods approach

PONE-D-24-27240R2

Dear Dr. Author,

We’re pleased to inform you that your manuscript has been judged scientifically suitable for publication and will be formally accepted for publication once it meets all outstanding technical requirements.

Kind regards,

Daniele Romanello

Academic Editor

PLOS ONE

Additional Editor Comments (optional):

Reviewers' comments:

Reviewer's Responses to Questions

**Comments to the Author**

1. If the authors have adequately addressed your comments raised in a previous round of review and you feel that this manuscript is now acceptable for publication, you may indicate that here to bypass the “Comments to the Author” section, enter your conflict of interest statement in the “Confidential to Editor” section, and submit your "Accept" recommendation.

Reviewer #2: All comments have been addressed

Reviewer #3: All comments have been addressed

2. Is the manuscript technically sound, and do the data support the conclusions?

Reviewer #2: Yes

Reviewer #3: Yes

3. Has the statistical analysis been performed appropriately and rigorously? 

Reviewer #2: Yes

Reviewer #3: Yes

4. Have the authors made all data underlying the findings in their manuscript fully available?

Reviewer #2: Yes

Reviewer #3: Yes

5. Is the manuscript presented in an intelligible fashion and written in standard English?

Reviewer #2: Yes

Reviewer #3: Yes

6. Review Comments to the Author

Reviewer #2: Thank you for giving the chance to review this manuscript . The authors were covered and addressed the comments and suggestions I raised. So I have no Any comments further. Thank you again.

Reviewer #3: (No Response)

7. PLOS authors have the option to publish the peer review history of their article (what does this mean? ). If published, this will include your full peer review and any attached files.

**Do you want your identity to be public for this peer review?** For information about this choice, including consent withdrawal, please see our Privacy Policy .

Reviewer #2: **Yes: ** Asmamaw Deguale Worku

Reviewer #3: No

---

## [Editor Report · Acceptance letter]

PONE-D-24-27240R2

PLOS ONE

Dear Dr. Tebeje,

I'm pleased to inform you that your manuscript has been deemed suitable for publication in PLOS ONE. Congratulations! Your manuscript is now being handed over to our production team.

Kind regards,

on behalf of

Dr. Daniele Romanello

Academic Editor

PLOS ONE